# Revealing the Gene Diversity and Candidate Gene Family for Adaption to Environment Depth in Leucosiid Crabs Comparing the Transcriptome Assembly

Yi-Jia Shih [1,2], Yin-Ming Yang [1], Si-Te Luo [3] and Jia-Ying Liu [1,2,*]

1 Fisheries College, Jimei University, Xiamen 361021, China
2 Fujian Provincial Key Laboratory of Marine Fishery Resources and Eco-Environment, Xiamen 361021, China
3 School of Life Sciences, Xiamen University, Xiamen 361005, China
* Correspondence: jiayingliu@jmu.edu.cn

**Abstract:** The study of gene diversity in interspecies contributes to our understanding of the environmental adaptation, evolutionary history, origin, and stability of biodiversity. Crabs are the main component of the macrobenthos community; therefore, how crabs adapt to different environments can be a guide to understand how to maintain community diversity. Previous studies on environmental adaptation have focused on differences in morphology, organ structure, and function, but there is a lack of research that explores this topic from the perspective of gene diversity in benthonic crabs. In this study, the leucosiid crab was selected for transcriptome assembly and was analyzed as this superfamily is one of the main clades in brachyuran crabs. Their transcriptome data were used to understand the gene diversity, phylogeny, and divergence time estimations. Subsequently, candidate gene families for depth adaptation were found in eight species that live in habitats with different depths. The results indicated that the number of total unigene sequences was between 65,617 (*Philyra malefactrix*) and 98,279 (*Arcania heptacantha*) in eight species. The total length of the genes was counted to be between 48,006,211 and 95,471,088 bp. The age of the superfamily Leucosioidea is over 150 Ma, dating back to at least the Jurassic geological period. The divergence of the family Leucosiidae would have occurred in the middle Cretaceous (around 100 Ma). After dividing into groups of three depth types, which carried the gene families, it was found that the three groups shared the solute carrier family, whereas bile acid secretion, organic solute transporter subunit alpha-like, and solute carrier organic anion transporter families only existed in the shallow group. This result shown that the gene function of ion concentration regulation might one of the candidate gene families related to the environmental adaptation of the leucosiid crab. Hence, these gene families will be analyzed in future studies to understand the mechanism of depth adaptation regulation in crabs.

**Keywords:** brachyura; Leucosioidea; transcriptome assembly; gene diversity

## 1. Introduction

Species are the constituent and basic evolutionary units of ecosystems, creating an important material basis [1]. Understanding what species constitutes and defines biodiversity is critical for managers and policy makers. Biodiversity is usually defined as the sum of genes, species, and ecosystems in a region, and is divided into four types, including taxonomic diversity [2], ecological diversity [3], morphological diversity (genetic diversity and molecular diversity) [4], and functional diversity [5]. When studying a community, understanding how many species a community consists of is crucial scientific evidence for identifying the genetic diversity of the species [6,7]. This can contribute to understanding the environmental adaptation of species, the competition between species, and the survival, reproduction, evolution, and stability of community biodiversity [6]. Genetic variability among individuals within a species may result from genetic recombination or mutation,

genetic polymorphism, the isolation of gene pools, local selection pressures, environmental complexity and gradients, and landscape mosaicism. Previous studies have relied on abundant species molecular databases and comparative analyses of genetic diversity, and their results have been used to develop the strategies and policies for biodiversity conservation [8–12].

Infraorder Brachyura, also called the "true-crab", is the largest and most diverse clade of the Decapoda group with over 7400 species [13]. These crabs contribute to the health and sustainability of ecosystems [14]. Due to the divergence of their morphology, organ structure and function, they can adapt to different habitats, increasing their habitat diversity [14,15]. Moreover, crabs are the main component of the macrobenthos community. Our previous studies also indicated that benthic crabs are one of the indicator species in intertidal regions, such as wetland and mangrove environments [3,16]. Therefore, the value of the biodiversity of crabs is important to study. Recently, research on the biodiversity conservation of crabs, community ecology, and species diversity have been focused on. The molecular data of crabs have been widely revealed in previous studies, including the sequences of single and short fragments, and mitochondrial genome and whole genome data; these data have been mainly applied to understand systematic and genetic mechanisms, physiological regulation, and genetic selection [17–24], but studies on gene diversity and the environmental adaptation mechanism between species or communities are relatively lacking. Notably, the regulation mechanism of how crabs change their habitat from sea to land is an important key to understanding the evolution of crabs. Traditionally, morphology and physiology have been explored to explain the habitat adaptation of crabs [25–27]. In recent years, several reports on the regulation of osmotic pressure on the gene level have also been published. According to the results, ion transporter gene families are one candidate gene of regulation during salinity transitions in sea water [28–30]. Therefore, the composition and expression of ion transporter gene families carried by marine organisms are valuable to use in the study of environmental adaptation.

The superfamily Leucosioidea is one of the key groups in the brachyuran clade; it includes 2 families, 3 subfamilies, 77 genera, and 504 species (WoRMS, https://www.marinespecies.org, accessed on 30 June 2022). Among them, 84 species were recorded in the gene bank, and although most of them had short-segment data, only 2 species had mitochondrial genomes (NCBI, https://www.ncbi.nlm.nih.gov/, accessed on 30 June 2022). Moreover, the habitat depth of leucosiid crabs ranges from the intertidal zone to over 500 m in the sea [31–33]. Therefore, a study on the gene function of leucosiid crabs is very suitable for exploring the regulation mechanism of osmotic pressure in crabs. The results can provide valuable evidence regarding the environmental adaptation mechanism and ecological service ability of marine invertebrates. However, leucosiid crabs have not attracted as much attention as more economically important species and the rarer species are not easy to investigate in particular. Previous studies focused on basic species identification and taxonomy. Furthermore, the whole genome or transcriptome is also lacking in the leucosiid group, which also limits the possibility of further studies on genetic function and environmental adaptation mechanisms.

Herein, the aim of this study is to reveal the gene diversity of leucosiid crabs and to clarify the systematic and estimated divergence time in these species. Subsequently, we also try to identify the composition members and the number of ion transporter gene families carried by different species of leucosiid crabs by comparing the transcriptome. The results will provide an important clue for the study of marine environment adaptation in crabs.

## 2. Materials and Methods

### 2.1. Sample Collection and RNA Preparation

In this study, eight species were selected for the first transcriptome assembly. The species included: *Iphiculus spongiosus*, *Arcania heptacantha*, *Myra celeris*, *Nursia plicata*, *Paranursia abbreviata*, *Philyra malefactrix*, *Pyrhila pisum*, and *Seulocia latirostrata*. They were all collected by trawler from Sanya, Hainan, China, on 29 August 2021. The fresh, live speci-

mens were bred under 25 °C seawater in the aquarium of the laboratory at the Fisheries College of Jimei University. In order to obtain more comprehensive transcriptome sequence information, the muscles, gills, liver, and heart were collected from each individual and placed into a tube containing liquid nitrogen for rapid freezing. All tissues were immediately stored in liquid nitrogen after sampling, then stored at −80 °C. Subsequently, total RNA was extracted from each sample by using TRIzol reagent (Invitrogen, Carlsbad, CA, USA) to obtain the transcriptome.

*2.2. Illumina RNA-Seq Library Preparation, Sequencing and Assembly*

For *A. heptacantha, M. celeris, N. plicata, P. abbreviata, P. malefactrix, P. pisum,* and *S. latirostrata*, next-generation sequencing was performed as the transcriptome sequencing method (Table 1). RNA isolation and sequencing were performed by BGI Genomics (BGI Genomics Co., Ltd., Shenzhen, China). RNA-seq libraries were prepared using the TruSeq mRNA Library Prep Kit v2 (Illumina, San Diego, CA, USA). DNA fragment length distributions were measured on the Agilent 5300 Fragment Analyzer (Agilent, Santa Clara, CA, USA) prior to pooling. Sequencing was performed on an Illumina NovaSeq 6000 Sequencing System (Illumina, USA) with the $2 \times 150$ base pairs and paired-end option. Before the de novo assembly process, the option to trim the adapter and low-quality reads was utilized using Trimmomatic v0.39. The de novo transcriptome assembly was then performed using Trinity version 2.8.4.

**Table 1.** Description of the data of eight species' transcriptome size.

| No. | Species | Method | Total Unigene Sequences | Total Length (bp) |
|---|---|---|---|---|
| 1 | *Iphiculus spongiosus* | PacBio SMRT sequencing | 73,809 | 56,483,844 |
| 2 | *Arcania heptacantha* | Next-Generation Sequencing | 98,279 | 95,471,088 |
| 3 | *Myra celeris* | Next-Generation Sequencing | 94,664 | 83,854,763 |
| 4 | *Nursia plicata* | Next-Generation Sequencing | 76,236 | 71,668,640 |
| 5 | *Paranursia abbreviata* | Next-Generation Sequencing | 85,700 | 52,623,429 |
| 6 | *Philyra malefactrix* | Next-Generation Sequencing | 65,617 | 48,006,211 |
| 7 | *Pyrhila pisum* | Next-Generation Sequencing | 73,709 | 53,931,646 |
| 8 | *Seulocia latirostrata* | Next-Generation Sequencing | 69,054 | 67,046,175 |

*2.3. PacBio cDNA Library Construction and Single-Molecule Real-Time (SMRT) Sequencing*

For *I. spongiosus*, the PacBio SMRT analysis was performed for transcriptome sequencing (Table 1). After RNA was extracted from all samples, 1 μg of RNA was used for PacBio cDNA library construction. We constructed one library from a mixture of the muscles, gills, liver, and heart. The SMRT sequencing library was prepared with the Clontech SMARTer PCR cDNA Synthesis Kit (Clontech Laboratories, Inc., Mountain View, CA, USA) and the BluePippin Size Selection System protocol. The mRNA enriched by Oligo (dT) magnetic beads was reverse transcribed into cDNA. Then, double-stranded cDNA was generated with the optimum cycle number. In addition, a >4 kb size selection was performed using the BluePippin Size Selection System. Next, large-scale PCR was performed for the subsequent SMRTbell library construction. The SMRTbell library was sequenced by BGI Genomics.

After PacBio sequencing was completed, the original disconnecting and low-quality reads were carried out. The output was filtered and processed with SMRTlink software (https://github.com/PacificBiosciences/pbbioconda, accessed on 29 September 2022), and the final data obtained were considered valid. Self-correcting subreads formed a CCS (circular consensus sequence) to obtain high-quality, transcriptional, and consistent sequences. The nonchimeric consistent sequences containing the 5′ primer, 3′ primer, and poly (A) tail are called full-length nonchimeric sequences (FLNCs). Since there are a large number of redundant sequences in the FLNCs, the redundant sequences need to be clustered together to remove the redundancy by the ICE algorithm and obtain corrected consensus sequences.

### 2.4. Sequencing Data Treatment and Annotation

CD-HIT was used for sequence alignment and clustering, and the corrected transcript sequences were clustered to remove redundancy according to the 95% similarity between the sequences. The assembled transcripts were annotated through several steps to identify coding regions within the transcripts and determine the homology search, protein family, and domain profiles using TransDecoder, BLAST+, HMMER, and PFAM, respectively. These annotation steps were wrapped up into a single pipeline with Trinotate. The annotation profiles were based on the UniProt database.

### 2.5. Analysis of Phylogenetic Cladogram and Divergence Time Estimation

For every Trinity "unigene", which is the gene unit in each of these assemblies, we retained the longest isoform for downstream analysis. Protein-coding genes were predicted from each single-isoform assembly using TransDecoder and validated by searching against the Swiss-Prot and Pfam databases, as noted above. Next, we used the combined set of predicted proteins to infer orthogroups using OrthoFinder. OrthoFinder was run with DIAMOND for local protein alignment, MAFFT to generate multiple sequence alignments, and IQ-TREE for gene tree inference. To build a phylogeny for all taxa, we used a subset of orthogroups, chosen by OrthoFinder, that consisted entirely of single-copy genes. These orthogroups were then concatenated and aligned across species with MAFFT. After determining the best-fitting protein substitution model for our data under the BIC criterion with ModelFinder, we generated the phylogenetic relationship based on a concatenated approach using IQ-TREE. Divergence times were estimated between 8 species based on the amino acid sequences of 99 single-copy orthologous genes shared by these species. A maximum likelihood tree was constructed by using RaxML with the GAMMA BLOSUM62 model. Divergence times were estimated by using MCMCtree of the PAML 4.0 package, using the approximate likelihood method with an independent substitution rate (clock = 2) and the GTR substitution model with $1.0 \times 10^7$ iterations and discarded $10^6$ iterations as the burn-in. The known time point information from Timetree (http://timetree.org/, accessed on 29 September 2022) was used in the divergence time estimation.

### 2.6. Analysis and Annotation of Ion Transporter Gene Families

The output of OrthoFinder was parsed to identify gene families. The species tree based on the concatenated alignments of single-copy orthologs was time-calibrated using the r8s v1.8.1 program with the penalized likelihood method. To provide node age constraints necessary for the time calibration, previously inferred point estimates of the divergence time between sister species from four different subsections were used. To assess the gene family evolution of the eight species, we only used the gene families with more than three gene copies per family and the species ultrametric tree as inputs to the CAFE (Computational Analysis of gene Family Evolution) tool.

## 3. Results and Discussion

### 3.1. The Results of the Eight Species' De Novo Transcriptome

The de novo transcriptome of some economic crabs was revealed by Illumina next generation sequencing [24,34,35]. Although next generation sequencing provides good technical support for the analysis of transcriptomes, the technology is still limited for short-read fragments. The short-read fragments could have assembly errors, which also affect the application of subsequent data [36,37]. SMRT sequencing technology is not limited by the read length; therefore, this technology could prevent any errors in transcriptome de novo assembly [37]. However, the analysis cost of SMRT sequencing is still high, and it requires the samples to be fresh; thus, it is not widely used in non-economic crabs. For this reason, the transcriptome size of eight species of leucosiid crabs was revealed for the first time in this study. These data can also provide a reference for transcriptome comparison and gene diversity in further studies.

The results of the sequencing method and transcriptome in each leucosiid species are shown in Table 1. The number of total unigene sequences was counted to be between 65,617 (*Philyra malefactrix*) and 98,279 (*Arcania heptacantha*) in the eight species. The total lengths of the genes were counted to be between 48,006,211 and 95,471,088 bp. SMRT sequencing identified through assembly that *I. spongiosus* did not carry the largest number of total unigene sequences, but rather *A. heptacantha* had largest number of total unigene sequences. This phenomenon may be caused by two reasons. One reason was the difference in the original carrying genes between interspecies, whereas the other reason may be due to errors in the short-fragment assembly [36,37]. This requires further investigation.

*3.2. Phylogenetic Cladogram and Divergence Time Estimation*

The divergence time of species and the geographical distribution of populations are important information for the formulation of biodiversity conservation strategies [38]. In this study, the creation of cladograms and the differentiation of ages of eight species were carried out by using the single-copy orthologous genes. The divergence time analysis implies that the age of the superfamily Leucosioidea is over 150 Ma, dating back to at least the Jurassic. Thus, the divergence of the family Leucosiidae might have occurred in the middle Cretaceous (around 100 Ma) (Figure 1). The earliest known fossil Brachyura is a part of the Dromiacea, which was dated from the late Jurassic [39], and Tseng et al. [15] proposed a hypothesis of late Jurassic to early Cretaceous origins of the brachyuran clade in some groups. The results of divergence time estimation in this study were highly concordant with those from previous studies.

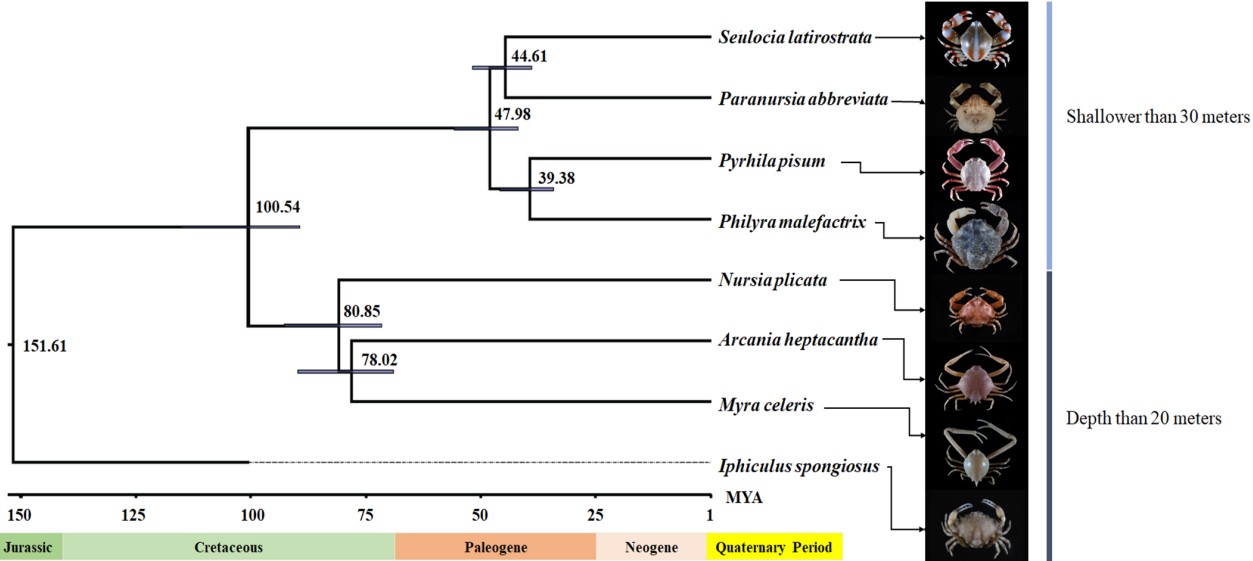

**Figure 1.** Phylogenetic cladogram and divergence time estimation of eight species in superfamily Leucosioidea.

*3.3. Ion Transporter Gene Families in Eight Leucosiid Species*

The process of species evolution is accompanied by the expansion and contraction of gene families, and this phenomenon also causes gene diversity among species [40]. In this study, eight leucosiid species were significantly differentiated by their habitat depths; therefore, the ion transporter gene families were analyzed to understand their adaption to environments of different depths. The results indicated that the recorded habitat depths of species could be divided into three groups: over 150 m, between 20 to 55 m, and less than 20 m (Figure 2). Through the selection and annotation of ion transporter gene families, the results show that the main gene families were different between these three groups of habitat depths. Furthermore, the number of gene members had similar trends related to habitat depth.

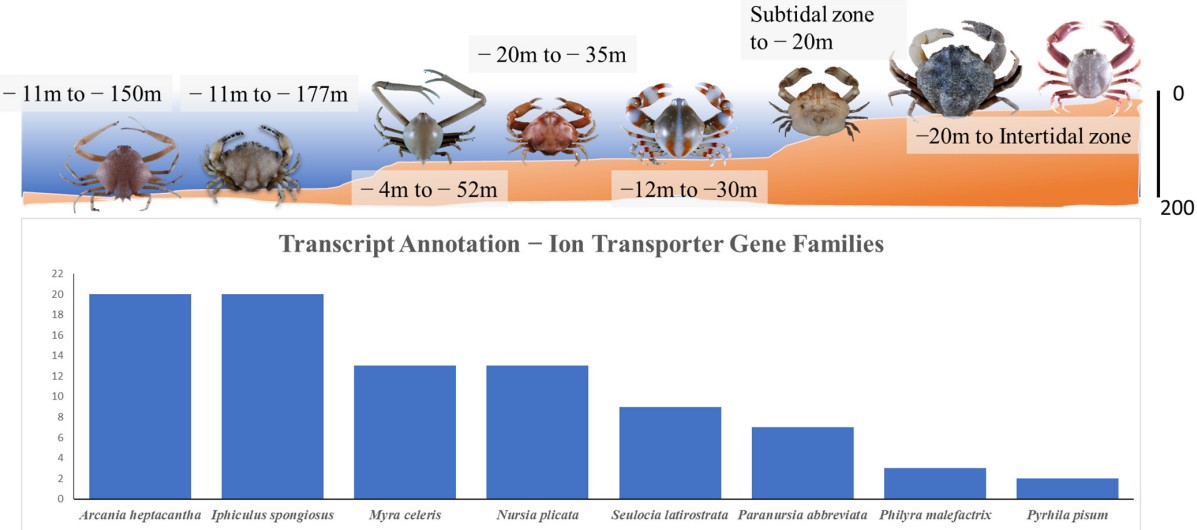

**Figure 2.** The relationship between depth distribution and number of ion transporter gene families in each species.

The deep group, which inhabits a depth over 150 m, carried up to 27 gene members, including SLC27A2, ndy-2, SLC10A2, SLC12A9, SLC16A13, SLC1A3, SLC22A4, SLC25A35, SLC25A40, SLC2A5, SLC30A6, SLC35A3, SLC35B3, SLC35C2, SLC35E1, SLC35F6, SLC37A2, SLC39A11, SLC39A9, SLC46A3, SLC4A1AP, SLC50A1, SLC6A14, SLCO4A1, TMEM184B, TMEM184C, and an unnamed gene. They belong to three gene families: the organic solute transporter ostalpha, solute carrier family, and transporter activity.

The middle group, which inhabits a depth around 20–50 m, carried 22 gene members, including SLC22A8, SLC31A1, SLC35A3, SLC35E1, SLC6A8, TMEM184B, TMEM184C, SLC10A6, SLC1A3, SLC25A25, SLC25A26, SLC25A35, SLC27A4, SLC30A6, SLC35B3, SLC35C2, SLC39A11, SLC46A3, SLC4A1AP, SLC6A12, SLCO5A1, and an unnamed gene. They belong to five gene families: organic solute transporter, organic solute transporter ostalpha, organic solute transporter subunit alpha-like, solute carrier family, and transporter activity.

The shallow group, which inhabits a depth that ranges from 20 m to the intertidal zone, carried a significantly lower number of genes, including Slc13a3, SLC22A8, SLC35A3, SLC35B3, SLC35B4, SLC35E1, SLC35F6, SLC51A, SLC6A8, and an unnamed gene. They belong to four gene families: bile acid secretion, solute carrier family, organic solute transporter subunit alpha-like, and solute carrier organic anion transporter family.

Environmental factors have been identified as limiting factors for species survival. In particular, the adaptation of marine organisms to salinity affects their geographic distribution [41,42]. The salinity of seawater varies significantly based on vertical depth. In general, the salinity of the marine surface and intertidal zone varies more dramatically than in the deep sea due to the tides, rainfall, river injection, and human influence [43]. Therefore, the crabs' response to osmotic pressure stress at different depths caused by salinity changes becomes a challenge when adapting to different environments. Previous studies indicated that the osmolyte contributes to osmotic regulation processes in invertebrates and responds to osmotic pressure through the accumulation of organic solutes [30]. Moreover, choline is a constituent of bile acid. It is considered a compatible osmolyte that maintains the balance of osmotic pressure under hypertonic stress in organisms [28,44,45].

The results show that the three groups of crabs all contain the solute carrier family. This family regulates SLC proteins on the cell surface. They are regarded as gatekeepers of the cell environment and dynamically respond to different metabolic states, including the regulation of Na$^+$ and K$^+$ [30]. This function of regulation might be used for seawater adaptation in crabs.

In addition, three gene families, the organic solute transporter subunit alpha-like, solute carrier organic anion transporter, and bile acid secretion families, only existed in the

shallow group. The first two gene families are responsible for the regulation of organic solute transporters [45], and the last one regulates the synthesis of bile acid [46]. The result might be caused by the species of the shallow group as they are more vulnerable to periodic salinity changes than those in the deep-sea group. Therefore, it is speculated that the shallow species might need more genes that regulate hyperosmotic stress to adapt to the various osmotic pressures for survival [47].

In a further analysis of the gene subfamilies, the number of gene members of the organic solute transporter ostalpha was the highest, and the solute carrier family, numbering 35, was second. The third subfamily was the solute carrier family, numbering 22, followed by the organic solute transporter subunit alpha-like (Figure 3). Among them, the function of organic solute transporter ostalpha has been described in previous studies. Their main mechanism consists of regulating the balance of the concentration of Na$^+$ and K$^+$ between the organism cell and its environment [48,49]. This shows that the function gene that regulates ion concentration has a candidate gene family for the environmental adaptation of leucosiid crabs. Hence, we will analyze these gene families in future studies to understand the mechanism of deep adaptation regulation in crabs.

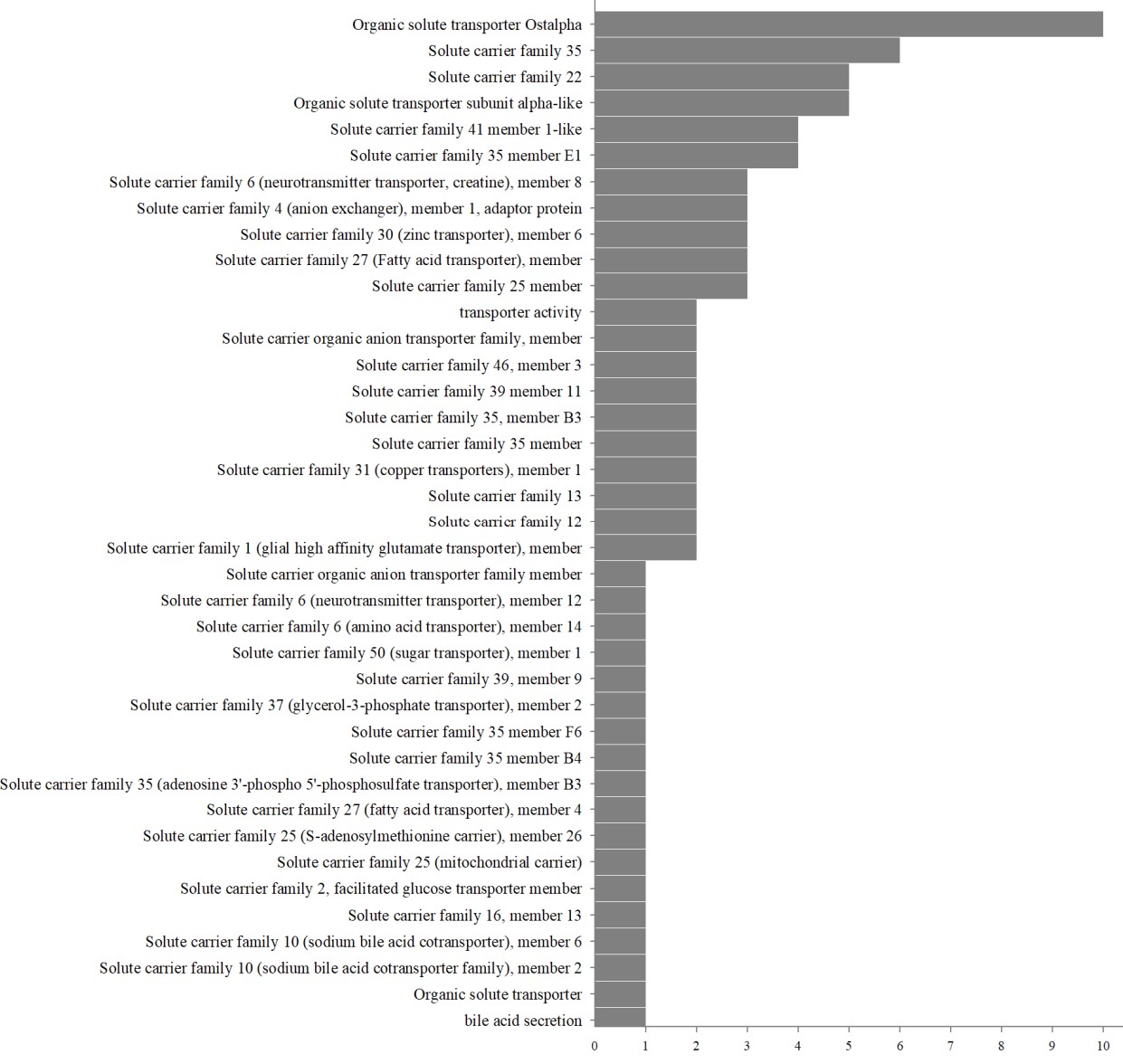

**Figure 3.** Function annotation and number of carried genes of ion transporter gene subfamilies in eight leucosiid species.

*3.4. Deposition of Samples and Transcript Data*

All the specimens were deposited at Fisheries College, Jimei University, China. The transcriptome assembly sequence was deposited in the NCBI Sequence Read Archive (SRA) database with accession numbers PRJNA818491 and PRJNA936839.

## 4. Conclusions

The transcriptome provided in this study could enrich the current genome database and help advance the studies of evolution and biodiversity. Here, eight de novo transcriptomes from leucosiid crabs were revealed, which can act as a reference for further transcriptome assembly in the leucosiid crab group. These data also serve to define the candidate gene families for environmental adaptation. The main results in this study identified the transcriptome size for each species, and the number of total unigene sequence was counted to be between 65,617 (*Philyra malefactrix*) and 98,279 (*Arcania heptacantha*). The age of the superfamily Leucosioidea was inferred to be over 150 Ma, dating back to at least the Jurassic. Thus, the divergence of the two leucosiid families occurred in the middle Cretaceous. Subsequently, based on the carried gene families in the groups of three depth types, it was identified that the solute carrier family was widely found in all species, whereas bile acid secretion, organic solute transporter subunit alpha-like, and solute carrier organic anion transporter families only existed in the shallow group. This result shown that the gene function of ion concentration regulation might one of the candidate gene families for the environmental adaptation of the leucosiid crab. In future studies, these gene families will be further analyzed to understand the mechanism of depth adaptation regulation and biodiversity formation for crabs.

**Author Contributions:** Conceptualization, Y.-J.S., Y.-M.Y. and J.-Y.L.; methodology, Y.-J.S. and S.-T.L.; software, Y.-J.S., Y.-M.Y. and S.-T.L.; validation, Y.-J.S., Y.-M.Y. and J.-Y.L.; formal analysis, Y.-J.S.; investigation, Y.-J.S. and Y.-M.Y.; resources, Y.-J.S. and Y.-M.Y.; data curation, Y.-J.S., Y.-M.Y. and S.-T.L.; writing—original draft preparation, Y.-J.S., Y.-M.Y. and S.-T.L.; writing—review and editing, Y.-J.S. and J.-Y.L.; visualization, Y.-J.S., Y.-M.Y., S.-T.L. and J.-Y.L.; supervision, Y.-J.S. and J.-Y.L.; project administration, Y.-J.S.; funding acquisition, Y.-J.S. All authors have read and agreed to the published version of the manuscript.

**Funding:** This work was supported by research grants from the National Science Foundation of Fujian Province (Grant No. 2020J05136), Distinguished Young Scientific Research Talents Plan in Universities of Fujian Province (Grant No. JAT190347), and National Natural Science Project Cultivation Fund of Jimei University (Grant No. ZP2020023). The funders had no role in the study design, data collection, and analysis, decision to publish, or preparation of the manuscript.

**Data Availability Statement:** Not applicable.

**Acknowledgments:** All authors thank BGI Genomics Co., Ltd. for help with the RNA extraction and third-generation sequencing, and XC Li, who works at Victory Time Co., Ltd., for the advice on bioinformatics analysis. We thank T.J. Chu for his contributions to the comments on the manuscript. We would also like to thank the anonymous reviewers, whose useful suggestions were incorporated into the manuscript.

**Conflicts of Interest:** The authors declare no conflict of interest.

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
