# Peer review of "Revealing the Gene Diversity and Candidate Gene Family for Adaption to Environment Depth in Leucosiid Crabs Comparing the Transcriptome Assembly"

_water, doi:10.3390/w15061246_

Round 1

Reviewer 1 Report

This study revealed a de novo transcriptome of I. spongiosus and provided an important resource for designing gene probes regarding target sequence capture. This finding could help to resolve the taxonomic system of the superfamily Leucosioidea. However, it is worth noting that there are many problems in the article. I have some suggestions for improvement as follows.

1.     Line 12: The word “link” should be modified into “linking”.

2.     Line 14-17: The sentence has a grammatical error. The word “though” was used incorrectly. There should be no “but” after it.

3.     Line 23: The words “enrich” and “develop” should be modified into “enriching” and “developing”.

4.     Line 48: Notice one usage: “contribute to doing sth”. And the spelling of the word “ exosystemic ” is wrong.

5.     Line 50-51: The word “adaption” should changed into “adapt”. The word “increases” should be modified to “increase”, the subject is plural. And the word “comprise” here should be replaced with a noun.

6.     Line 61-63: The word “ systematic ” is unmatched. The word “ including ” is used incorrectly. And change "group" into "groups".

7.     Line 71: The spelling of the word “ ptrygostome ” is wrong.

8.     Line 94: The word “was” does not match the subject.

9.     Line 99: The word “ grab ” is used incorrectly.

10.  Line 123: There is a extra letter “f” at the end of this row.

11.  Line 156: Use "were" after "read".

12.  Line 165: The word “ achieve ” is used incorrectly.

13.  Line 201: The first letter of the word “total” should be uppercase.

Author Response

Thanks for your conceding to review our study. All reviewers gave valuable comments and suggestions to help improve the paper. According to all comments, we considered the direction of the journal and the common opinions of reviewers, we have major revised this article. We combined 7 de novo transcriptomes of leucosiid crab and compared. Try to understand to the gene diversity, phylogeny and divergence time estimations. Subsequently, candidate gene families for depth adaptation were found from eight species which different depth of habitats. The revision of MS that follow by reviewer reported were listed in attached files. Thanks for your attention.

Reviewer 2 Report

Dear Editors,

Dear Authors,

The study entitled: “Revealing the genetic-diversity of lecosiid crab: a de novo transcriptome assembly of Iphiculus spongiosus Adams & White, 1849 (Decapoda; Brachyura; Leucosioidea)” represents valuable study that deserves to be published. Substantive content of the manuscript is correct, however, sometimes it is difficult to evaluate. At all, the manuscript requires major revision. The title of the manuscript does not correspond to the study aim. The introduction chapter is inconsistent and does not satisfactorily introduces a reader to the study subject. Materials and methods are not sufficiently described. The data presentation should be also improved. In some places, the manuscript has been written imprecisely, and a lot of terms and sentences require clarification. Language presentation is low and requires significant improvement. I recommend professional language correction.

In conclusion, I do not recommend the reviewed manuscript in the present form for possible publication in the Water periodical. Authors should make corrections and try again after major revision. All remarks, questions and fixes were placed in the attached pdf file (yellow highlights contain fixes and sentence suggestions, while red highlights contain comments and questions).

Thank you for another interesting manuscript that I could review!

Author Response

(The authors gave the same response as above.)

Reviewer 3 Report

The manuscript by Shih et al. entitled " Revealing the genetic-diversity of lecosiid crab: a de novo transcriptome assembly of Iphiculus spongiosus Adams & White, 1849 (Decapoda; Brachyura; Leucosioidea)". The authors only have described a de novo transcriptome assembly. I found there are some mistakes as well as concept misunderstandings in this manuscript, despite the authors spending tremendous efforts to perform functional annotation of transcriptome assembly in I. spongiosus. The title is inappropriate. The authors did not reveal the genetic-diversity of lecosiid crab. The abstract is dreadful and looks as though it has been written by an inexperienced team member, without careful scrutiny by the rest of the team. This will need to be rewritten in order to achieve clarity and give the reader the confidence that the manuscript is going to be worth reading. In the introduction section, the author must have revised and described how the use of the de novo model transcriptome in phylogenetic analysis. Authors must describe examples of real applications. The author describes a lot of parts about the phylogeny of Heterotremata in the Introduction section. But this part doesn't seem to matter with the results and discussion. Results and discussion section, the narrative of the results and discussion is too simple like a running account. Discussion must be improved by removing general statements and explaining the results obtained and comparing these results with genetic studies of I. spongiosus. Further, I think the paper may be more suitable for a more specific journal such as BMC genomic data. In conclusion, the MS will likely require pretty large amounts of changes, lying between a deep re-arrangement and an almost complete re-writing. The manuscript will not be accepted for publication until editing of the English grammar and phrasing by a native English speaker. Unfortunately, while I consider most sections of the article very poorly organized in their current form, in my opinion, this manuscript does not meet the criteria for publication and must therefore be rejected.

I have no minor comments.

Author Response

(The authors gave the same response as above.)

Reviewer 4 Report

This paper mainly aimed to report and characterize the RNA sequences of Iphiculus spongiosus. Frankly the title of MS did not reflect what is report in the paper. I see nothing about genetic diversity. Only three sample can be enough to explore genetic diversity of this species. Also wondering how this work contribute to taxonomy and diversity of Brachyura. Also I see nothing in discussion of this work with other Brachyura works or relevant species. Authors need to rewrite carefully and try to reconsider about what is the main contribution of this paper in evolution of Brachyura. If you just report the structure and assembly of RNA for this species, Water is not suitable for this work at all. 

Also, author need to ask English native speaker to check the writing. I would lot of flaws, typo and grammatical error. 

Good Luck 

Author Response

(The authors gave the same response as above.)

Round 2

Reviewer 2 Report

Dear Editors,

Dear Authors,

The study entitled: “Revealing the gene-diversity and the candidate gene family for adaptation of environmental depth in lecosiid crab: a case by compared the transcriptome assembly of eight species” has been improved but, in my opinion, there is still a lot to do. Substantive content of the manuscript seems to be correct, however, the described material and methods are inconsistent which makes it difficult to evaluate the manuscript's full correctness. The number of species varies from 7 to 9 across subchapters. Two methods of transcriptome sequencing were used but no information has been placed why. Some software was not included in the M&M chapter, e.g., BEAST software. Moreover, no data concerning the motivation for each analysis has been placed, which makes it difficult for the reader to understand and follow the logical structure of methods applied. The data presentation should be also improved as many low-quality pictures are placed. Overall language quality of the manuscript is very poor, and a lot of grammar and style mistakes are present, which discourages the reader from further reading the manuscript. In my last review, I have recommended for the professional language correction. This time I also keep my recommendation because there is not significant improvement in the language presentation of the manuscript.

In conclusion, I do not recommend the reviewed manuscript in the present form for possible publication in the Water periodical. Authors should make corrections and try again after major revision. All remarks, questions and fixes were placed in the attached pdf file.

Thank you!

Author Response

Thanks for your conceding to review our study. All reviewers gave valuable comments and suggestions to help improve the paper. According to all comments, we considered the direction of the journal and the common opinions of reviewers, we have major revised this article followed the comment from reviewers. The title was also minor change to: Revealing the gene-diversity and the candidate gene family for adaption of environmental depth in lecosiid crab: a case by compared the transcriptome assembly. We will send this MS to the professional language editing for the English writing error. Other revision of MS that follow by reviewer reported were listed in attached files, respectively.

Reviewer 4 Report

It looks better than the previous one, but still need more work to improve the manuscript, particularly English writing. I found lot of mistakes. Frankly, such flaws downgrade the work, even the content is relatively interesting. Please carefully check and send to native English speaker to recheck. 

Also I think "Water" is probably not suitable for your work. Your work is more likely genetic analyses and diversity. "Gene" or "Diversity" seem to match well with the paper content. 

Title:

- What is the "environmental depth"? Why not "depth environments" ? Any better wording available? 

- Really need to emphasize "eight species" ? 

Abstract:

- What is " stability of community biodiversity" ? I think the first sentence of abstract is a bit work. Please brush up and shorten it. 

Introduction:

- What did you means "the aim of this study was first transcriptome assembly for 8 species" ? Please rewrite. 

- "Depending the data to carry out the analysis of clade diagram and the time of species divergence" Don't understand what did mean for ? and it is not a sentence! 

- "...the function annotation of gene families to explored the candidate..." Must be "explore"

Materials and Methods:

- Did you really sequences for eight species ? I think I remember that you did only one species for previous version? I

- Scientific name must be italic (Line 91-92).

- Why Pacbio for Iphiculus and apply different method to other species? Is it possible for you to compare data from different methods? 

- Did you apply codon partition into each gene when running phylogeny in IQTree? 

Don't you thinking about include outgroup into your analysis ???

Results & Discussions:

- " In previou studie" ??? What does it mean ??

- Ion transporter gene family should be mentioned briefly in "Introduction" Why does this come to your focus? How this relate to depth adaptation.

- Figure 2, deep distribution or depth distribution ???

- Can you expand Figure 3? It is difficult to see. 

- I don't understand how dating analysis related to your findings in depth adaptation. Please elaborate?

- " Discussed to the three groups of inhabit in different depth were carried gene families, the results indicated that the three group shared the solute carrier family, while bile acid secretion, organic solute transporter subunit alpha-like and solute carrier organic anion transporter family were only existence in shallow group" - I think this is the same as in abstract. I expect to see more detail and explanation in "Discussion part"

Cheers

Author Response

(The authors gave the same response as above.)

Round 3

Reviewer 2 Report

Dear Editors,

Dear Authors,

The manuscript entitled: “Revealing the genetic-diversity of lecosiid crab: a de novo transcriptome assembly of Iphiculus spongiosus” has been significantly improved. All my remarks and questions were fully addressed. Language presentation has been well improved, but the manuscript draft requires some minor improvements (typos, etc.).

Best regards,

Author Response

We are much grateful for your careful reading of our manuscript and your valuable suggestions. We reviewed the article carefully and revised the spell error.
